# Tryptophan and Substance Abuse: Mechanisms and Impact

**DOI:** 10.3390/ijms24032737

**Published:** 2023-02-01

**Authors:** Majid Davidson, Niloufar Rashidi, Md Kamal Hossain, Ali Raza, Kulmira Nurgali, Vasso Apostolopoulos

**Affiliations:** 1Institute for Health and Sport, Victoria University, Melbourne, VIC 3021, Australia; 2Regenerative Medicine and Stem Cells Program, Australian Institute of Musculoskeletal Science (AIMSS), Melbourne, VIC 3021, Australia; 3Fiona Elsey Cancer Research Institute, Ballarat, VIC 3353, Australia; 4Federation University, Ballarat, VIC 3353, Australia; 5Department of Medicine Western Health, Faculty of Medicine, Dentistry and Health Sciences, University of Melbourne, Melbourne, VIC 3021, Australia; 6Immunology Program, Australian Institute of Musculoskeletal Science (AIMSS), Melbourne, VIC 3021, Australia

**Keywords:** tryptophan, kynurenine, serotonin, addiction, mental health, methamphetamine

## Abstract

Addiction, the continuous misuse of addictive material, causes long-term dysfunction in the neurological system. It substantially affects the control strength of reward, memory, and motivation. Addictive substances (alcohol, marijuana, caffeine, heroin, methamphetamine (METH), and nicotine) are highly active central nervous stimulants. Addiction leads to severe health issues, including cardiovascular diseases, serious infections, and pulmonary/dental diseases. Drug dependence may result in unfavorable cognitive impairments that can continue during abstinence and negatively influence recovery performance. Although addiction is a critical global health challenge with numerous consequences and complications, currently, there are no efficient options for treating drug addiction, particularly METH. Currently, novel treatment approaches such as psychological contingency management, cognitive behavioral therapy, and motivational enhancement strategies are of great interest. Herein, we evaluate the devastating impacts of different addictive substances/drugs on users′ mental health and the role of tryptophan in alleviating unfavorable side effects. The tryptophan metabolites in the mammalian brain and their potential to treat compulsive abuse of addictive substances are investigated by assessing the functional effects of addictive substances on tryptophan. Future perspectives on developing promising modalities to treat addiction and the role of tryptophan and its metabolites to alleviate drug dependency are discussed.

## 1. Introduction

Drug addiction is identified as a compulsive relapsing disorder that has devastating effects on the brain and behavior of users and results in a disability to control the drug-seeking tendency and use, notwithstanding unfavorable consequences [1]. DA is initiated via experimental utilization of a recreational drug in society or exposure to a prescribed medicine from a friend [2,3]. Substances including methamphetamine (METH), alcohol, marijuana, heroin, and nicotine are the most commonly used addictive drugs that may have disparate detrimental side effects on users. Numerous short-term and long-term side effects such as insomnia, drowsiness, increased blood pressure, uncontrolled movement, and rapid heart rate are the dire and unfavorable consequences of disparate addictive drug abuse [4,5].

In recent decades, alcohol addiction has been a global crisis that accounts for almost 5% of the disease burden [6]. Based on the statistics, alcohol is the most prevalently abused substance in the United States and is responsible for 3.5% of annual deaths due to its detrimental and deleterious effects, such as liver cirrhosis and neurotoxicity [7]. Therefore, developing cost-effective and promising drugs to treat alcohol addiction is crucial, as only a small proportion of patients can achieve effective remission. Marijuana (whose street names include dope, pot, grass, weed, and hashish) manifests as an addictive drug extracted from the cannabis plant. Tetrahydrocannabinol is the active ingredient in marijuana, which is the main reason for its potential for dependency. This addictive substance can significantly change perception due to its psychoactive characteristics. Depending on the mode of administration, marijuana can produce various neuropsychological disorders such as hallucinations, abnormal happiness, and difficulty concentrating [8]. Heroin, another commonly used addictive substance produced from morphine [9,10], is highly abused among people from various social cultures. Heroin targets the brain and receptors on cells, which increases the heart rate and causes breathing and sleeping issues [11,12]. In addition, dry mouth, drowsiness, constipation, and depression are common side effects [13].

The World Health Organization predicts that almost one-third of the global adult population smokes tobacco, accounting for many smoking-related deaths [14,15]. Nicotine (one of the foremost effective ingredients in tobacco) is a naturally manufactured alkaloid used as an anxiolytic stimulant [16]. Similar to other psychostimulant agents, nicotine increases extracellular dopamine concentrations in the nucleus accumbens, which can cause neuropsychological/behavioral alterations in users [17]. Table 1 provides information regarding common addictive substances.

Despite the availability of current treatment options, such as behavioral therapy and medication-assisted treatment, they are often ineffective for addressing the underlying mechanisms of addiction [37,38]. These treatments typically involve behavioral therapies, medications, and support groups, but they do not address the underlying neurobiological mechanisms of addiction. Understanding the mechanisms of addiction and its physiological impacts can provide an opportunity for developing a more efficient treatment for this global health issue. Tryptophan is an essential amino acid that plays vital roles in several critical biological processes, including the regulation of mood and behavior [39]. Addiction leads to an imbalance in the levels of tryptophan and its metabolites, which can result in serious health issues [40]. Understanding the connection between tryptophan and addiction may hold potential for developing novel therapeutic approaches for addiction treatment.

In this review article, the detrimental effects of addictive substances, specifically METH, on human mental and psychological dependency are evaluated. The role of METH and other addictive substances on mental and psychological dependency in humans is presented, along with a description of tryptophan metabolites in the mammalian brain and their potential as new treatment options for controlling compulsive substance abuse. The functional effects of METH on tryptophan and its disparate metabolites, including serotonin 5-HT (5-hydroxytryptophan), melatonin, kynurenine, and reactive oxygen and nitrogen species, as well as the formation and structure of each, are examined. The understanding of the connection between tryptophan metabolites and addiction provides a valuable opportunity to investigate future perspectives on the development of promising modalities for addiction treatment.

## 2. General Effects of Addiction on the Body

The abuser of disparate types of addictive materials usually suffers from one or more health-related challenges (Figure 1). Tobacco smoke may increase the risk of cancers, heart failure, brain stroke, lung diseases, and chronic obstructive pulmonary diseases such as emphysema and bronchitis. Smoking also increases the risk of tuberculosis, certain eye diseases, and autoimmune system diseases such as rheumatoid arthritis [41,42].

Marijuana is known as the most commonly abused addictive drug in the United States, and its use is expanding substantially in all adult age groups of both sexes. Abusing these addictive substances can significantly enhance the risk of permanent intelligence quotient (IQ) loss by eight points. Moreover, different studies have corroborated the existence of a close relationship between marijuana abuse and psychotic disorders such as depression, anxiety, and suicidal ideation [43,44].

Likewise, the unusual use of alcohol is increasing worldwide and is regarded as one of the main reasons for chronic liver diseases [45]. Long-term abuse of alcohol may also increase the risk of chronic diseases such as hypertension, decreased immune system functioning, digestive problems, and cancer.

Caffeine, a stimulant found in coffee, tea, energy drinks, and various medications, acts on the central nervous system by binding to adenosine receptors, leading to increased alertness and decreased fatigue [46]. While moderate caffeine consumption may provide some health benefits, excessive caffeine intake can lead to various adverse health outcomes, such as insomnia, anxiety, high blood pressure, and an increased risk of certain types of cancer [47]. Long-term heavy caffeine use can also lead to tolerance, withdrawal symptoms, and difficulty reducing or stopping caffeine consumption, ultimately leading to addiction.

Cocaine is one of the most addictive substances, and its abuse may cause the emergence of many unfavorable and dangerous side effects such as mood disorders, cardiovascular diseases, weight loss, an increase in the potential risk of human immunodeficiency virus infection, and memory loss [48].

METH is an amine drug that is abused due to its sympathomimetic effects and is taken via inhalation, ingestion, injection, or smoking [49,50]. METH is a weak lipophilic base that diffuses across the plasmalemma of the presynaptic neurons and vesicular membranes [51]. METH enters the monoaminergic terminals of neurons due to its structural similarities to monoamine hormones (serotonin, adrenaline, noradrenaline, dopamine, and melatonin). It then accumulates in the synaptic vesicles through the action of vesicular monoamine transporters. It causes an initial outward flow of hormones along with the prevention of the reuptake of neurotransmitters and their metabolism by enzymes such as monoamine oxidase [52,53,54]. Following the initial METH abuse, an increase in the levels of monoamines occurs [51], leading to an elevation in intra-synaptic levels of monoamine [55]. This increase results in hallucinations, euphoria, and anorexia [49], as well as risk-taking sexual behaviors, which raises serious public health and safety concerns [56]. Repetitive and escalating doses of METH result in decreased monoamine levels due to a reduction in monoamine transporter binding sites as well as in the activity of synthetic enzymes [57]. The abuse of addictive drugs such as METH confuses addicts on whether reward-specific circuits or adaptive natural reward circuits are activated in their brains. As such, addicted individuals feel that drug abuse is part of their natural biological needs [58]. However, studies have shown that multiple high-dose administrations of METH result in drug resistance in the body, a phenomenon known as drug tolerance [59]. Levels of the tryptophan metabolite, which is an amine precursor to serotonin, melatonin, kynurenine, and quinolinine pathways, have been noted to be affected by METH addiction [60]. Decreased levels of tryptophan have also been found to take place in suicidal adolescent patients [61]. Clinically, it is known that METH toxicity affects nearly every organ system in the body, with the most considerable damage occurring to the CNS [62]. This includes metabolic compromise, oxidative stress, and inflammation of neurons resulting in neuronal death [63].

## 3. Tryptophan and Its Metabolites

Tryptophan is an important amino acid that plays a crucial role in various biological processes, including in an infant′s growth, muscle development, enzyme function, and neurotransmitter regulation [64,65]. Tryptophan is a precursor to the metabolites serotonin, melatonin, and kynurenine [66]. Most of the daily intake of tryptophan oxidizes down the kynurenine pathway, whilst the rest degrades via serotonin pathways [67]. Tryptophan is metabolized into serotonin and melatonin via the generation of 5-hydroxytryptophan, which includes human peripheral tryptophan hydroxylase 1 (hpTrpH1) or human neural tryptophan hydroxylase 2 (hnTrpH2), depending on whether metabolism occurs in peripheral entero-chromatic cells or neural cells of the CNS. 5-hydroxytryptophan is then metabolized into serotonin and, further down, into melatonin via the action of different enzymes [66,67]. The kynurenine pathway either involves the generation of kynurenic acid (KA) or the formation of 3-hydroxy-anthranillic acid, further leading to quinolinic acid (QA). These components, QA and KA, are believed to have noteworthy impacts on CNS neurons. QA possesses the potential for application as a neurotoxin, while KA is a neuroprotectant. 3-hydroxykynurenine is the third kynurenine metabolite that may create free radicals and aggravate neuronal damage [68].

The kynurenine pathway plays a critical role in regulating immune responses and has the potential to be a target for various inflammatory and neurological disorders by modulating the immune response and regulating the production of kynurenine derivatives [69]. Due to tryptophan catabolism′s incredible impact on increasing immune suppression, disparate small-molecule inhibitors have been developed and tested in clinical trials [70,71,72]. Inflammatory cytokines and chemokines are key mediators of the immune response, and their release can lead to a cascade of effects that can disturb the immune system and cause a corresponding increase in kynurenine production [73,74]. An imbalance in kynurenine production can shift the balance of tryptophan metabolism, resulting in the accumulation of pro-inflammatory and neurotoxic kynurenine derivatives [75]. This can lead to a range of negative effects, including increased susceptibility to infections, chronic inflammation, and neurodegeneration. Furthermore, research has shown that the kynurenine pathway is closely linked to various psychiatric and neurological disorders, including depression, anxiety, schizophrenia, and Alzheimer′s disease [76].

The kynurenine pathway involves 95% of tryptophan degradation in the liver via two rate-limiting enzymes: indoleamine 2,3-dioxygenase (IDO) and tryptophan 2,3-deoxygenase (TDO) [77]. These enzymes, TDO and IDO, control tryptophan degradation through the kynurenine pathway. TDO is primarily expressed in the liver and is responsible for the initial degradation of tryptophan. This enzyme converts tryptophan into kynurenine, which is then further metabolized by IDO. IDO is expressed in various tissues, including the immune cells, and plays a key role in the regulation of immune responses. TDO usually leads to nicotinamide dinucleotide (NAD+) in the liver only. However, IDO is known to carry the same extra-hepatical reaction but in much smaller amounts than the enzyme TDO [67]. Additionally, the combination of IDO1 inhibitors together with immune checkpoint inhibitors, such as anti-programmed cell death 1 (PD1) and anti-programmed cell death ligand 1 (PD-L1), have opened new avenues towards increasing the efficacy of immuno-oncology in cancer treatment [78,79,80].

The mammalian brain contains from nanomolar to low micromolar concentrations of kynurenine pathway metabolites [81]. Acute stress causes an increase in tryptophan metabolism to kynurenine in mice brains and plasma [82]. In fact, increased IDO enzyme activity and IdoI-v1 expression have been noted in the hypothalamus and frontal cortex of the brain, specifically in astrocytes. Ferulic acid, a plant-derived phenol compound with anti-inflammatory effects, has been shown to have suppressing effects on lipopolysaccharide-induced IDO messenger RNA (mRNA) expression in mouse microglial cells [83]. In addition, ferulic acid suppresses the phosphorylation of p38 MAPK and the nuclear translocation of NF-κB [83]. Figure 2 schematically illustrates the tryptophan metabolites, their pathways, and their effects on the CNS.

## 4. Tryptophan and Addiction: METH Addiction as an Example

Tryptophan is an essential amino acid that can only be obtained via dietary intake, and a diet with low levels of tryptophan is related to malnutrition or poor diet habits [67]. Short-term and long-term abuse of alcohol and its associated withdrawal is closely related to tryptophan metabolism/disposition in humans and animals. Various studies have proved that the activity of the rate-limiting enzyme of tryptophan degradation (liver tryptophan pyrrolase) increases substantially during acute alcohol abuse and during withdrawal [84]. The occurrence of this alteration causes a significant change in the synthesis process of brain serotonin and the appropriate availability of tryptophan to the brain [84]. It has been reported that serotonin can be considered the prominent tryptophan metabolite related to psychiatric comorbidity in abstinent cocaine-addicted patients [40]. However, the non-existence of efficacious treatments following the great relapse rate among cocaine abusers remains a major health problem.

METH (under the street names ice, crystal, glass, or kryptonite) is a potent synthetic central nervous system (CNS) drug that has been placed second in rank of application in the United States, Asia, and Oceania after cannabis [85,86]. METH is often sold in powder or crystal form and is administered via intranasal sniffing, oral ingestion, pulmonary inhalation, and injection. METH addiction leads to serious side effects on disparate organs within the human body, such as subjective euphoria/arousal, chronic depression, and psychomotor activation [87,88]. Poor mental health and social disability (i.e., unemployment and economic hardship) can be considered the most immediate adverse effects of METH dependence, which significantly decreases the quality of life among users [89,90]. METH dependence is a major global public health challenge due to its medical, psychiatric, cognitive, and socioeconomic consequences [91,92]. Figure 3 represents the METH-induced serotonin pathway in the CNS.

Weekly stimulant use of recreational drugs such as METH, cocaine, etc., results in decreased levels of tryptophan. Suicidal adolescent patients suffering from major depressive disorder have decreased levels of circulating tryptophan [61], and suicidality/depression is considered one of the symptoms of METH addiction. Overall, research has shown that METH abuse affects the CNS, the spleen, the gut, and the liver [60]. Damage to these organ systems can also adversely affect tryptophan metabolism pathways, causing an imbalance in the levels of its metabolites. The effect of METH on tryptophan and its metabolites and the associated consequences are briefly discussed below.

### 4.1. Serotonin (5-hydroxytryptophan)

Hydroxylation of L-tryptophan leads to the formation of L-5-hydroxytryptophan (L-5HT) via the action of tryptophan hydroxylase. Aromatic amino acid de-carboxylate helps convert L-5HT to serotonin 5-HT. Tryptophan hydroxylase 1 (TPH1) is known to be composed of two types of TPH enzymes, and is responsible for 95% of the body’s 5-HT formation in the gut, which can travel to all body organs via the blood except for the brain, as it cannot pass through the blood–brain barrier. TPH2 helps the formation of 5-HT in the brain along with the action of aromatic l-amino acid decarboxylase (AADC) [93]. It has been shown that tryptophan hydroxylase activity increases in the globus pallidus, nucleus accumbens, and the caudate area within 1 h of a single bout of METH. However, long-term multiple METH doses only decreased TPH2 activity in the caudate area but not in the nucleus accumbens and globus pallidus [94]. However, recently, it was noted that TPH2 gene variants are not a factor in vulnerability to METH psychosis/dependence as no genotypic/allelic changes are observed between METH-dependent patients (n = 162) and controls (n = 243) [95].

Further experiments would aid in the understanding of why there is a lack of correlation between long-term and short-term METH use around the activity of TPH2 in the nucleus accumbens and globus pallidus regions of the brain. Despite this, a decrease in TPH2 levels is observed in the caudate area in both short- and long-term METH administration. These studies show that the decrease in TPH2 activity occurs in different brain regions, and it is plausible to argue that a decrease in TPH2 levels in the brain could be responsible for low 5-HT levels. It is essential to keep in mind that other factors/molecular mechanisms could also be responsible for low 5-HT levels in the CNS, such as tolerance-related 5-HT deficits in the brain [59].

### 4.2. Melatonin

Melatonin is a metabolite of serotonin that is formed via the serotonin pathway through the action of enzymes aralkylamine N-acetyltransferase (AA-NAT) and hydroxyindole-O-methyltransferase (HIOMT). After its formation in the pinealocytes of the pineal gland, it is transferred to the blood and cerebrospinal fluid of the brain, where second messengers of sleep are initiated in order to cause sleepiness [67]. Melatonin is formed through the metabolism of serotonin. Therefore, a decrement in serotonin levels would mean a decrease in melatonin levels in METH addicts [57,96]. As such, serotonin is also known to regulate sleep–wake cycles [67], which would explain the disturbance in the sleep cycles of METH addicts. Overall, amphetamine abuse is known to be linked to disturbances in sleep cycles, with chronic abuse being more detrimental than the usual acute abuse of drug addiction [97]. A study in rhesus macaques showed disrupted wake cycles and poor sleep efficiency after both low and high METH self-administration [98].

### 4.3. Kynurenine

The metabolism of tryptophan forms kynurenine via the action of the enzymes IDO and TDO [61]. Kynurenine is not known to be neuroactive but can cross the blood–brain barrier to aid in the formation of free radicals producing 3-hydroxyanthranillic acid, 3-hydroxykynurenine, or quinolinic acid. It has also been shown that depressed, suicidal patients have decreased kynurenine plasma levels [61], and suicidality is one of the major symptoms of METH addiction. Pro-inflammatory cytokines are known to further help kynurenine metabolism and cause an increase in the levels of its metabolites [66]. Alterations in the evel of the kynurenine pathway metabolites are known as a major cause of neurotoxicity [99]. Over-activation of the IDO and TDO enzymes is known to cause the degradation of most of the tryptophan along the kynurenine pathway. Thus, this leads to lower levels of serotonin via the upregulation of tryptophan along the kynurenine pathway, which is thought to be one of the reasons for the occurrence of depression [99].

### 4.4. Kynurenic Acid

The action of kynurenine aminotransferase on kynurenine leads to the formation of kynurenic acid, which takes place in the neurons, astrocytes, and oligodendrocytes of the CNS. Once it is synthesized, it is released into the extracellular spaces to act on its pre- and post-synaptic targets [67]. Physiologically, kynurenic acid has a neuroprotective role as it helps prevent pro-inflammatory cytokines, but its elevated levels are known to be linked to psychosis and cognitive deficits [66]. It also inhibits the alpha-7 nicotinic receptor because the receptor, when activated by its agonist, is known to have cognitive-enhancing effects [100]. Kynurenic acid is also known to stabilize the levels of free radicals within the CNS [66]. D-amphetamine, an enantiomer of amphetamine, when systematically administered to rats of different ages, causes a decrease in the extracellular levels and tissue content of kynurenic acid in the brain within 1 h following D-amphetamine administration [101]. Schizophrenic patients with psychosis and a history of suicide have lower cerebrospinal fluid kynurenic acid concentrations [102]. However, patients with schizophrenia only have relatively higher levels of kynurenic acid levels in cerebrospinal fluid when compared with healthy individuals [66]. This shows that depression/suicidality can lead to lower levels of kynurenic acid. Kynurenic acid is known to act as an antagonist for glutamatergic NMDA receptors. Trihexyphenidyl is known to antagonize the METH-induced reward pathway via the suppression of dopamine release from the mesolimbic area [103]. Present data show that high levels of kynurenic acid can prove to be beneficial in the CNS of METH addicts, but kynurenic acid is found to be low in suicidal depressive and psychotic patients, which are common major symptoms of METH addiction.

### 4.5. Xanthurenic Acid

The catabolism of the 3-hydroxykynurenine forms xanthurenic acid via the action of kynurenine aminotransferase. It is also considered an analog of kynurenic acid and is thus also known to have neuroprotective functions. It is thought to be localized in the cell body and dendrites of neurons but not in the axons, which is why there are debates on whether xanthurenic acid has a role in neurotransmission in the CNS [104]. However, the upregulation of xanthurenic acid in the CNS leads to a decreased synthesis of neurotransmitters such as serotonin and dopamine [105]. In addition, levels of depression and levels of xanthurenic acid positively correlate with each other in the urine of patients with depression [106]. Some initial studies also show that xanthurenic acid may act as an allosteric agonist for the metabotropic glutamate receptors type 2 and 3, but further research is required to support this finding. Along with xanthurenic acid, they are known to act as inhibitors for vesicular glutamatergic transporters whose immediate action is the prevention of glutamate reuptake from the synaptic vessels [107]. A similar mechanism takes place via METH in the neural synapses of a METH addict′s CNS neurons. The current data show that while xanthurenic acid may have neuroprotective characteristics, its upregulation has similar effects as those seen in METH addicts. Therefore, future research on xanthurenic acid in the context of METH addiction will aid in a better understanding of the neural actions of xanthurenic acid in the CNS.

### 4.6. 3-hydroxyanthranillic Acid

In the CNS, 3-hydroxyanthranillic acid is formed via the action of kynureninase on anthranilic acid, whereas in the periphery, it is formed from 3-hydroxykynurenine [66]. 3-hydroxyanthranillic acid is a very reactive compound that can either act as an antioxidant or pro-oxidant depending on the redox reactions in the brain. One study showed no notable differences in the plasma levels of 3-hydroxyanthranillic acid between suicidal depressed and healthy control groups [61]. However, a positive correlation between the levels of 3-hydroxyanthranillic acid and choline were noted; choline is responsible for cell membrane breakdown in melancholic depressive patients [66]. Inhibition of kynurenine in the kynurenic acid pathway leads to increased production of 3-hydroxyanthranillic acid [108], and since depressive patients have low levels of kynurenic acid in their brains [102], it is possible that depressive patients have high levels of 3-hydroxyanthranillic acid. Based on this, it can be hypothesized that depression can be caused by METH abuse, and 3-hydroxyanthranillic acid levels would be higher in depressive METH addicts. Further research, however, is required to better understand the relationships between METH addiction, depression, and 3-hydroxyanthranillic acid.

### 4.7. Quinolinic Acid

Quinolinic acid is formed from 3-hydroxyanthranillic acid via the action of enzyme 3-hydroxyanthranillic acid oxygenase. Quinolinic acid is mainly thought to be produced in the brain by microglial cells and by infiltrating macrophages and is thought to be neurotoxic via several different mechanisms [66]. Quinolinic acid is known to severely impair energy metabolism in the striatum of developing rats via the activation of NMDA receptors [109]. Quinolinic acid has also been known to cause an increase in the release of glutamate while preventing its reuptake from the synaptic vesicles in individuals [110]. Quinolinic levels have been found to be very high in the cerebrospinal fluid of depressed suicide attempters [111]. High levels of quinolinic acid correlate with interleukin (IL)-6, suggesting that high levels of quinolinic acid may be due to inflammatory responses [112]. In addition, quinolinic acid levels are high in the cortical and subcortical regions of psychiatric suicidal patients [113]. In fact, ketamine, an NMDA receptor antagonist, helps overcome suicidal thoughts [114]. However, even though ketamine can act as an antagonist for the NMDA receptor, a balanced ratio of quinolinic acid/kynurenic acid can be more effective in avoiding depression with respect to the NMDA receptor because both the acids can act as an agonist (quinolinic acid) and antagonist (kynurenic acid) for the receptor [111]. D-amphetamine, an enantiomer of amphetamine, is known to cause a decrease in levels of kynurenic acid [101]. As such, it is plausible to argue that a decrease in kynurenic acid would push tryptophan metabolism towards quinolinic acid and, thus, raise the concentration of quinolinic acid, which would support the fact that quinolinic acid is one of the factors responsible for depression/suicidality in METH addicts.

### 4.8. Picolinic Acid

When the enzyme 2-amino-3-carboxymuconic-6-semialdehyde decarboxylase (ACMSD) acts on ACMS, rather than forming quinolinic acid spontaneously, it instead forms picolinic acid [66]. It is not known whether picolinic acid is able to cross the blood–brain barrier or not, but it can be easily detected in human cerebrospinal fluid [115].

ACMSD is also present in glial and neuronal cells of the cortex and hippocampus [113]; hence, picolinic acid can be produced in the brain. A study showed that picolinic acid can antagonize the neurotoxic effects of quinolinic acid, as well as having neuroprotective properties [116]. In depressed suicide attempters, picolinic acid levels in the cerebrospinal fluid are known to significantly drop when compared with the levels in healthy individuals, which is linked to a drop in the actions of the ACMSD enzyme [117]. This means that an increase in quinolinic acid levels will occur because 2-amino-3-carboxymuconic-6-semialdehyde spontaneously converts to quinolinic acid in the absence of the ACMSD enzyme. Based on these findings, it can be concluded that lower levels of picolinic acid can be expected in the brains and bodies of people with METH addiction. However, the relationship between picolinic acid and METH addiction is not fully known, and further studies will provide a better understanding of this relationship.

### 4.9. Nicotinamide Adenine Dinucleotide (NAD^+^)

Nicotinamide adenine dinucleotide (NAD^+^) is formed from the metabolism of quinolinic acid via the action of quinolinic acid phosphoribosyl transferase [61]. The TDO enzyme of the tryptophan pathway, which is known to carry out most of the metabolism in the tryptophan pathway in the liver, also leads to the formation of NAD^+^. It is also considered to have a significant role in the maintenance of cell viability [67]. In one study in mice, which were forced to swim, it was noted that NAD^+^ had anti-depressant effects in mice after they were stressed [118]. NAD^+^ is also found to be an important co-enzyme in many oxidation–reduction reactions [67]. It is also responsible for cell homeostasis and energy metabolism, thus making it an important cellular molecule that is a regulator of the circadian cycle [119]. NAD^+^ can have positive effects in METH addicts who suffer from suicidal depression, but further research is required to explore the correlation between NAD^+^ and METH-induced neuronal damage.

### 4.10. Reactive Oxygen and Nitrogen Species

Reactive oxygen (ROS) and nitrogen species (RNS) are produced as by-products of the tryptophan metabolism pathway. Increases in ROS and RNS are known to be responsible for oxidative stress and nitrosative stress in the body, and this oxidative stress has been linked to anxiety, depression, and inflammatory responses. RNS and ROS usually include superoxide, nitric oxide, and hydrogen peroxide [120]. Based on this, increases in levels of RNS and ROS are very likely in the CNSs of METH addicts. It has also been noted that decreased levels of the antioxidative compound glutathione also occur in victims of major depressive disorder [121]. Due to this, antidepressants that target the production of RNS and ROS are currently being explored, which includes inhibiting the over-production of RNS and ROS along with suppressing their inflammatory effects [122]. METH affects the balance of tryptophan and its metabolites significantly, and further research on each of the pathways will be constructive towards the development of new therapeutics, which will aid in evading the detrimental effects of METH on the tryptophan metabolism pathways.

## 5. Tryptophan Metabolites and Treatment Options for Addiction

Acute/chronic addiction to alcohol has significant impacts on tryptophan metabolism in human and animal subjects. Tryptophan has been studied in relation to alcohol addiction because it is the precursor of indolylamine serotonin, which controls disparate functions in the CNS such as mood and depression [123,124]. It was corroborated in the mid-1980s that 5-HT had the ability to control the abnormal tendency to drink alcohol [84]. In animal studies, the presence of low central 5-HT concentration in various animal models was an interesting aspect of tryptophan metabolism related to alcohol addiction [125]. This deficiency is of great importance in alcohol-abuse behavior. The increasing trend of cocaine abuse has been a major health issue associated with millions of people′s dependency and high social costs [126,127]. Numerous studies have investigated the mechanism of action of cocaine, with no efficacious treatments for patients and compulsive users [128]. A small amount of tryptophan is used for 5-HT synthesis, while a greater proportion is metabolized via the kynurenine pathway manufacturing biologically active metabolites such as kynurenic acid, 3-hydroxykynurenine, and quinolinic acid. While 3-hydroxykynurenine and quinolinic acid are neurotoxic metabolites, kynurenic acid is thought to be involved in neuroprotection. Due to the great influence of 5-HT in the pathogenesis of mood disorders, the relationship between neurotoxic and neuroprotective metabolites of tryptophan is of great importance [129]. Currently, the pharmacological increase in kynurenic acid has been under extensive investigation as a new and promising option for treating those patients suffering from compulsive abuse of marijuana and nicotine dependency [40,130,131]. It has been corroborated that kynurenic acid can play an important role in relapse-like situations owing to its enhancement of an abrogating cocaine-seeking demeanor [131]. Table 2 presents information regarding the most commonly used addictive substances and their associated tryptophan metabolites.

## 6. Conclusions

This review evaluated the detrimental effects of addictive substances, specifically METH, on human mental and psychological dependency. The role of METH and other addictive substances in mental and psychological dependency in humans was presented, along with a description of tryptophan metabolites in the mammalian brain and their potential as new treatment options for controlling compulsive substance abuse. Tobacco smoke significantly increases the potential hazard of different chronic diseases such as lung diseases, heart failure, brain stroke, and chronic obstructive pulmonary diseases. The abuse of marijuana may result in various unfavorable side effects, such as the risk of permanent IQ loss, depression, and suicide planning. In the last five years, vaping has become popular, especially among the youth, and health issues and severe complications have already been reported; one awaits the addictive behaviors and health issues due to its use in the coming years. In the case of alcohol, it is worth mentioning that the long-term abuse of this addictive substance may cause the emergence of disparate chronic diseases such as high blood pressure, a decrease in the performance of the immune system, and digestive problems. Despite detrimental long-term results related to METH and other addictive substances, numerous people use them to alleviate short-term physical/emotional sensations. By eliminating these unfavorable states, drug-use behaviors are negatively reinforced. Abstinence from METH may cause a return to former undesirable emotional states that are associated with craving, usually leading to an elevated probability of relapse. The functional effects of METH on tryptophan and its disparate metabolites, including serotonin 5-HT (5-hydroxytryptophan), melatonin, kynurenine, and reactive oxygen and nitrogen species, as well as the formation and structure of each, were examined. Understanding the connections between tryptophan metabolites and addiction provides a valuable opportunity to investigate future perspectives on developing promising modalities for addiction treatment. The compulsive abuse of addictive substances as active CNS stimulants eventuates thse emergence of serious side effects such as poor mental/physical health and socio-economic challenges. Tryptophan metabolites show potential for application as new treatment options to manage compulsive abuse of addictive substances. Moreover, several studies have reported the development of vaccines against METH addiction [142,143,144], but none have been translated into human clinical trials yet. The identification of tryptophan metabolites as potential treatment options for addiction, in combination with new approaches such as vaccines, may lead to more effective methods for managing compulsive substance abuse in the future.

## Figures and Tables

**Figure 1 ijms-24-02737-f001:**
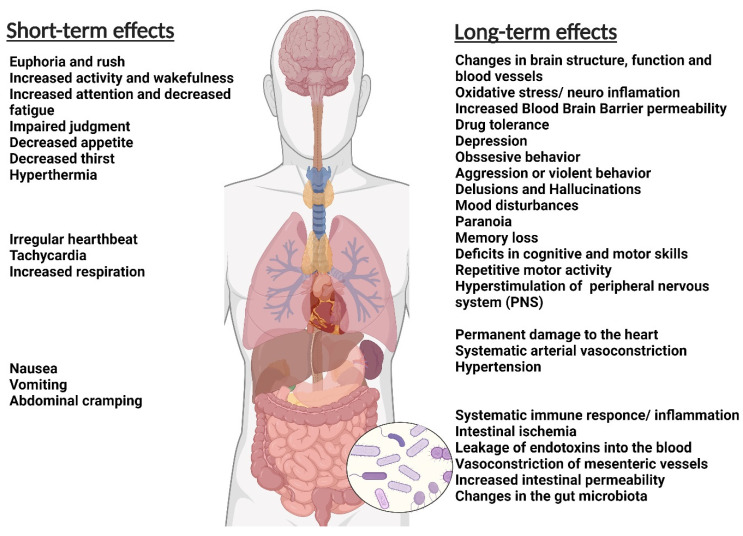
The general short-term and long-term effects of addiction on the body (Figure created with BioRender.com).

**Figure 2 ijms-24-02737-f002:**
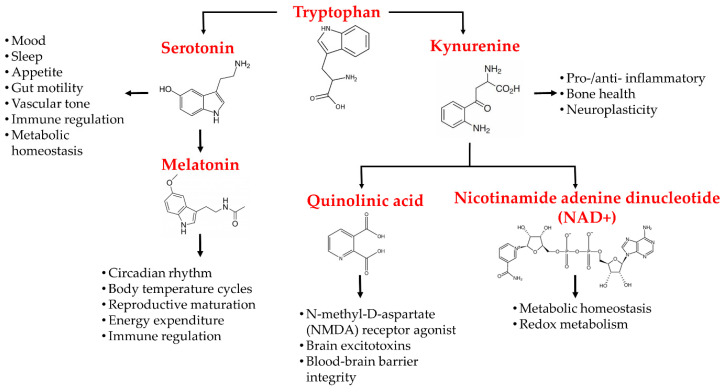
Schematic demonstration of tryptophan metabolites, its related disparate pathways, and its metabolites′ effects on the central nervous system.

**Figure 3 ijms-24-02737-f003:**
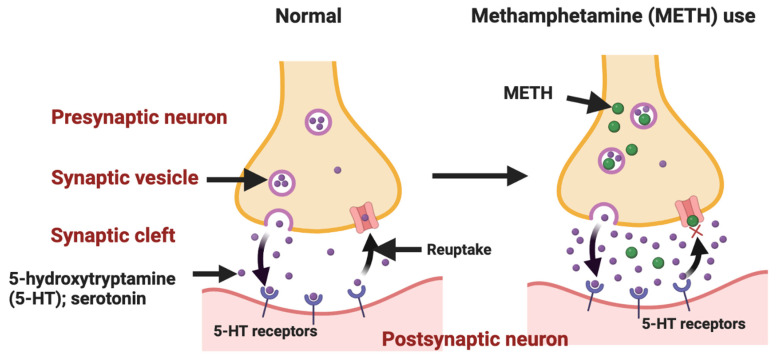
Methamphetamine (METH) administration leads to a detrimental effect on the serotonin pathway in the central nervous system [Figure created with BioRender.com].

**Table 1 ijms-24-02737-t001:** Some of the common addictive substances, mechanisms of action, and side effects.

Drug	Mechanism of Action	Chemical Structure	Mode of Administration	Side Effects	Ref.
Alcohol	Promotion of inhibitory neurotransmission	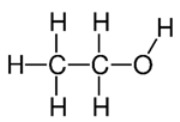	Oral	Liver cirrhosis Neurotoxicity Drowsiness Loss of consciousness Gaps in memory	[6,18,19]
Caffeine	Release of dopamine, serotonin, and noradrenaline	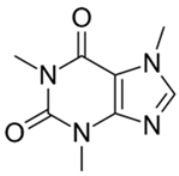	Oral	Increased vigilance and metabolic activity Insomnia Nervousness Restlessness Nausea	[20,21,22,23]
Cocaine	Potentiation of dopamine and norepinephrine	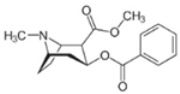	Injection Inhalation	Euphoria Increased concentration Pupillary dilation	[24,25]
Heroin	Stimulation of opioid receptor	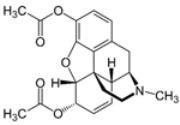	Injection Inhalation	Analgesia Depression Pupillary constriction Pneumonia Increased heart rate Difficulty sleeping Difficulty breathing Dry mouth	[10,26,27]
Marijuana	Stimulation of cannabinoid receptor type 1 and cannabinoid receptor type 2	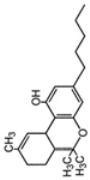	Oral Smoke	Tachycardia High blood pressure Decrease in saccadic accuracy Difficulty concentrating Hallucinations Abnormal happiness Depression	[28,29,30]
Methamphetamine	Catecholamine potentiation	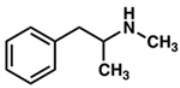	Injection Inhalation Oral Intranasal sniffing	Depression Euphoria Pupillary dilation Cardiovascular diseases	[31,32,33]
Nicotine	Nicotinic acetylcholine agonist	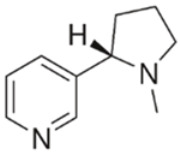	Oral Smoke	Delayed coronary healing Hypertension Tachycardia Smoke-associated dryness	[34,35,36]

**Table 2 ijms-24-02737-t002:** Summary of common addictive substances and their associated tryptophan metabolites.

Drug	Tryptophan Metabolite	Ref.
METH	5-HT, KA, and QA	[132,133]
Alcohol	5-HT, KA, 3-HK, and 3-HAA	[84,123]
Marijuana	IDO 1, KA, and QA	[134,135]
Heroin	QA and 5-HIAA	[136]
Nicotine	KA, 5-HT, KYN, HKY, and 3-HAA	[137]
Caffeine	KA, 5-HT, 5-HIAA, and QA	[138,139]
Cocaine	KA, KYN, 5-HT, and 5-HIAA	[140,141]

Abbreviations: IDO, indoleamine 2,3-deoxygenase; KA, kynurenic acid; KYN, kynurenine; L-5HT, L-5-hydroxytryptophan; METH, methamphetamine; QA, Quinolinic acid; 3-HAA, 3-hydroxyanthranillic acid; 5-HIAA, 5-hydroxyindoleacetic acid.

## Data Availability

No new data were created or analyzed in this study. Data sharing is not applicable to this article.

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
