# Peer review of "Tryptophan and Substance Abuse: Mechanisms and Impact"

_ijms, 2023, doi:10.3390/ijms24032737_

Round 1

Reviewer 1 Report

1.    Title of table 2 is absent

2.    The information contained in lines 446-448 is not a conclusion: “Several metabolites of METH 446 including serotonin 5-HT, melatonin, kynurenine, kynurenic acid, xanthurenic acid, 3-hydroxyanthranillic acid, quilinic acid, picolinic acid, NAD+, ROS, and RNS, and the formation structures are discussed”

3.    The conclusion section is a results summary, please improve this section. For instance, describe trends, perspectives, etc.

4.    The title of the work does not correspond to its content

5.    In general, the content of the manuscript should be reorganized to the title reflects its content. Some sections are not relevant.

Author Response

Reviewer 1:

 Title of table 2 is absent

Author’s response:

The authors appreciate your feedback. The authors have provided the required information in the manuscript (Line: 418).

  1. The information contained in lines 446-448 is not a conclusion: “Several metabolites of METH including serotonin 5-HT, melatonin, kynurenine, kynurenic acid, xanthurenic acid, 3-hydroxyanthranillic acid, quilinic acid, picolinic acid, NAD+, ROS, and RNS, and the formation structures are discussed”

Author’s response:

The authors appreciate your feedback. We have revised the text accordingly which we hope it improves the quality of the manuscript.

  1. The conclusion section is a results summary, please improve this section. For instance, describe trends, perspectives, etc.

 Author’s response:

The authors appreciate your feedback. The authors have provided the required information in the manuscript (line: 464-494).

  1. The title of the work does not correspond to its content

 Author’s response:

The authors appreciate your feedback. The title of review has been changed to address this comment.

  1. In general, the content of the manuscript should be reorganized to the title reflects its content. Some sections are not relevant.

 Author’s response:

The authors appreciate your feedback. The title of review have been changed accordingly which we hope it could improve the quality of manuscript.

Reviewer 2 Report

1- Authors should expand by writing on the role of Tryptophan metabolism, especially on two rate-limiting enzymes; tryptophan 2,3-dioxygenase (TDO) and indoleamine 2,3-dioxygenase (IDO) that  control TRP degradation through the KYN pathway.

2- Authors can also give a hint on disturbing the immune systems as a result of the rise of inflammatory cytokines and chemokines and its relation to the kynurenine (KYN) pathway

3- Inappropriate style of writing references

for example ref 2 :West, R.; Brown, J., Theory of addiction. 2013.

Author Response

Reviewer 2:

1- Authors should expand by writing on the role of Tryptophan metabolism, especially on two rate-limiting enzymes; tryptophan 2,3-dioxygenase (TDO) and indoleamine 2,3-dioxygenase (IDO) that  control TRP degradation through the KYN pathway.

Author’s response:

The authors appreciate your feedback and comment. Authors have added further information according your constructive comments which we hope it could improve the quality of the manuscript (Line: 155-207).

2- Authors can also give a hint on disturbing the immune systems as a result of the rise of inflammatory cytokines and chemokines and its relation to the kynurenine (KYN) pathway.

Author’s response:

The authors appreciate your feedback and comment. We have added further information according your constructive comments which we hope it could improve the quality of the manuscript (Line: 155-207).

3- Inappropriate style of writing references

for example ref 2 :West, R.; Brown, J., Theory of addiction. 2013.

Author’s response:

The authors appreciate your feedback and comment. The reference section have been revised which we hope it could improve the quality of paper.

Reviewer 3 Report

In this review, the authors aimed to discuss the role of tryptophan in alleviating the effects of different addictive substances on mental disorders, above all focusing on methamphetamine (METH).

In my opinion, although this is an interesting review, several major revisions need to be addressed before publication in International Journal of Molecular Sciences.

-        Acronyms should be better checked and introduced the first time a word appears in the text.

-        In general, although the authors mentioned different addictive substances, their focus is clearly on METH. For instance, in paragraph “1. Introduction”, a section is dedicated only to METH. Why did the authors focus only on METH and not on other addictive substances? I suggest restructuring the review describing the effects of all different addictive drugs/compound on tryptophan metabolites or specifying in the title but also in the Introduction that the current review highlights only the effects of METH on tryptophan.

-        Although the aim of the review was to discuss the potential role of tryptophan metabolites as new treatment options to control compulsive abuse of addictive substances, the review seems to be more focused on the effects of additive substances, above all METH, on tryptophan metabolites’ alterations.

-        In paragraph “4.2 Melatonin”, the authors reported that a study in rhesus macaques showed disrupted wake cycles and poor sleep efficiency after both low and high METH self-administration. However, this referenced study needs more details. For example, were melatonin levels analyzed?

-        In paragraph “6. Conclusions”, the last sentence “Recently, a promising approach was shown to induce immunity in animal models [155-157], which could potentially be used alone or in combination with tryptophan metabolite drugs” needs more explanations.

Author Response

Reviewer 3:

In this review, the authors aimed to discuss the role of tryptophan in alleviating the effects of different addictive substances on mental disorders, above all focusing on methamphetamine (METH).

In my opinion, although this is an interesting review, several major revisions need to be addressed before publication in International Journal of Molecular Sciences.

  • Comment: “Acronyms should be better checked and introduced the first time a word appears in the text.

Author’s response:

The authors appreciate your feedback. We have revised the text to address this comment.

  • Comment: “In general, although the authors mentioned different addictive substances, their focus is clearly on METH. For instance, in paragraph “1. Introduction”, a section is dedicated only to METH. Why did the authors focus only on METH and not on other addictive substances? I suggest restructuring the review describing the effects of all different addictive drugs/compound on tryptophan metabolites or specifying in the title but also in the Introduction that the current review highlights only the effects of METH on tryptophan.

Author’s response:

The authors appreciate your constructive comment. Authors have revised the title and introduction to address this comment. We hope this could improve the quality of this review (line: 73-95).

  • Comment: “Although the aim of the review was to discuss the potential role of tryptophan metabolites as new treatment options to control compulsive abuse of addictive substances, the review seems to be more focused on the effects of additive substances, above all METH, on tryptophan metabolites’ alterations”

Author’s response:

The authors appreciate your constructive comment. Authors have revised the manuscript to address this comment. We hope this could improve the quality of this review

  • Comment: “In paragraph “4.2 Melatonin”, the authors reported that a study in rhesus macaques showed disrupted wake cycles and poor sleep efficiency after both low and high METH self-administration. However, this referenced study needs more details. For example, were melatonin levels analyzed?”

Author’s response:

The authors appreciate your great feedback. Study by Andersen et al. (Reference 99) evaluated the effects of methamphetamine self-administration and abstinence on sleep-like measures derived from Actiwatch monitors in rhesus monkeys under well-controlled experimental conditions. Results of study showed that methamphetamine administration markedly disrupted sleep-like measures.

  • Comment: “In paragraph “6. Conclusions”, the last sentence “Recently, a promising approach was shown to induce immunity in animal models [155-157], which could potentially be used alone or in combination with tryptophan metabolite drugs” needs more explanations

Author’s response:

The authors appreciate your constructive comment. Authors revised the conclusion section, which we hope it could improve the quality of this review (line: 462-494).

Round 2

Reviewer 1 Report

No one

Reviewer 3 Report

The authors have addressed all my comments.